# Deep Learning with Physics Priors as Generalized Regularizers

**Frank Liu**
School of Data Science
Old Dominion University
Norfolk, VA 23508
United States

**Agniva Chowdhury**
Computer Science and Math Division
Oak Ridge National Lab
Oak Ridge, TN 37831
United States

## Abstract

In various scientific and engineering applications, there is typically an approximate model of the underlying complex system, even though it contains both aleatoric and epistemic uncertainties. In this paper, we present a principled method to incorporate these approximate models as physics priors in modeling, to prevent overfitting and enhancing the generalization capabilities of the trained models. Utilizing the structural risk minimization (SRM) inductive principle pioneered by Vapnik, this approach structures the physics priors into generalized regularizers. The experimental results demonstrate that our method achieves up to two orders of magnitude of improvement in testing accuracy.

## 1 Introduction

Learning from observation data is a crucial task in scientific machine learning (SciML). Deep neural networks have demonstrated to be highly effective in modeling complex systems in scientific research fields such as physics[4, 9, 19], chemistry[13, 31, 1] and biology[12, 5]. To avoid overfitting and to improve generalization performance, regularization techniques such as L1 and L2 regularization[29], weight decay[6], dropout[25], batch normalization[10] and early stopping[16] are commonly deployed during the training of models.

In many scientific machine learning applications, it is quite often that an approximate mechanistic model of the underlying physical phenomenon is available, albeit with uncertainty. For example, the motion in mechanical systems is governed by Newton's Law, and can be mathematically described by Hamiltonian mechanics[2], while the motion of fluid is governed by Navier–Stokes equations[27], and the electric and magnetic fields are described by Maxwell equations[11]. The concept of integrating physics models with more expressive neural network models was initially introduced decades ago[17, 20, 28]. More recent work among this "hybrid-model" or "grey-box" approach include [32, 26, 28, 18, 14, 3, 23, 15, 8, 21]. Among them, a notable work is [32], which proposed a method to integrate simplified or imperfect physics models with deep learning models, by combining the two models as additive right-hand-sides of the differential equations, while with focused applications on the trajectory forecasting of complex systems.

In this paper, we present the analysis that integrating information of a physics model to deep grey-box modeling can be achieved as a generalized regularizer. Recognizing that the simplified or imprecise physics model is subject to both aleatoric (from data) and epistemic (from model) uncertainties, we use Vapnik's structural risk minimization[30] as the inductive principle to cast the generalized regularization as an optimization problem.

NeurIPS 2023 AI for Science Workshop.

**Problem Description**

The observation data are represented by the tuple $\{(\hat{x}_j, \hat{y}_j)\}$ with $j = 1 \ldots, N_d$. The neural network model is denoted by $\mathbf{G}_w(\cdot)$ with $w$ representing trainable weights. By applying the L2 norm, the training of the model is equivalent to the empirical risk minimization(ERM) task below:

$$\mathcal{L}_d = \frac{1}{N_d} \sum_{j}^{N_d} (\hat{y}_j - \mathbf{G}_w(\hat{x}_j))^2 \tag{1}$$

When modeling the dynamic behavior of the physical system defined on $[0, T] \times D \rightarrow \mathbb{R}^m$, without loss of generality, we include time as part of the input. Hence $\hat{x}_j \in [0, T] \times D$ and $\hat{y}_j \in \mathbb{R}^m$. As in many scientific and engineering applications, we assume $\hat{y}_j$ is subject to noise.

To avoid overfitting, Vapnik introduced the concept of structural risk in the seminal work of [30]. The purpose of including structural risk is to prevent the model from becoming too complex. By penalizing models with large complexity under a given measure of the structural risk (e.g., VC-dimension) in the training process, the structural risk minimization (SRM) ensures that the models would not become too complex. The minimization is often realized on a sequence of nested structures (or hypotheses) with increasing structural risk:

$$\cdots \subset S_{k-1} \subset S_k \subset S_{k+1} \subset \cdots \tag{2}$$

with the recognition that more complex models with larger risks produce lower training loss (in the form of empirical risk), but with the increased potential to overfit. A learned model with a balanced trade-off between the empirical risk and structural risk will most likely to achieve good accuracy and generalization performance.

A widely adopted SRM in deep learning is the weight decay[6], where the ERM is augmented by a regularizer, which measures the L2 norm of the weights:

$$\mathcal{L}_{wd} = \frac{1}{N_d} \sum_{j}^{N_d} (\hat{y}_j - \mathbf{G}_w(\hat{x}_j))^2 + \lambda \|w\|^2 \tag{3}$$

with $\lambda$ as a hyper-parameter controlling the balance between the empirical and structural risk.

The motivation of our work is that in many SciML applications, it is quite common that a physics model is available. However the physics model is only an approximate of the underlying physical phenomenon of the complex system (otherwise we can simply use the mechanistic model itself). The uncertainties of the mechanistic model with respect to the underlying complex system include aleatoric uncertainties and epistemic uncertainties[24]. The former (or "data uncertainty") represents the inherent variability in the data, while the latter (or "model uncertainty") represents the imperfect model, which can due to the missing components in the model, or our lack of understanding of the physical phenomenon.

We propose to utilize the information embedded in the approximate model in model training by structuring the mechanistic model as a generalized regularizer. However unlike common L2-norm regularizer, the parameter $\lambda$ has stronger dependency on the disparity (or the empirical representation of the epistemic uncertainty) of the physics prior, and should be optimized in a more comprehensive way. Our method provides a different perspective to the deep grey-box modeling approach such as [32]. The introduction of the generalized regularization also opens the door to other interesting means of modeling training, such as the inclusion of multiple mechanistic models as physics priors with multiple regularizers, and the co-optimization of mechanistic model coefficients along with the model itself.

## 2 Physics Prior as Generalized Regularization

We refer to the (approximate) mechanistic model as the physic prior, it has the same support as the observation data $[0, T] \times D \rightarrow u$:

$$\mathcal{F}_\theta(u)(x) = 0 \tag{4}$$

where $u \in \mathbb{R}^m$. In the case where the physics prior is an ODE, proper initial condition should be specified $u(0) = u_0$. In the case the physics prior is a PDE, additional boundary condition is needed: $u(x_b) = u_b$ where $x_b \in \partial D$.

To include physics prior as a regularizer, we add additional collocation points in the support $\{x_i\}$ where $i = 1, \cdots, N_p$ and $x_i \in [0, T] \times D$. We introduce a generalized regularization as:

$$\mathcal{L}_p = \frac{1}{N_p} \sum_i^{N_p} \left(\mathcal{F}_\theta(u)(x_i)\right)^2 = \frac{1}{N_p} \sum_i^{N_p} \left(\mathcal{F}_\theta(\mathbf{G}_w(x_i)\right)^2 \tag{5}$$

The total loss is the combination of empirical loss in Eqn. 1 and Eqn. 5 [1]:

$$\mathcal{L}\left(\hat{x}_j, \hat{y}_j, x_i; w\right) = \mathcal{L}_d + \lambda \mathcal{L}_p = \frac{1}{N_d} \sum_j^{N_d} (\hat{y}_j - \mathbf{G}_w(\hat{x}_j))^2 + \frac{\lambda}{N_p} \sum_i^{N_p} \left(\mathcal{F}_\theta(\mathbf{G}_w(x_i)\right)^2 \tag{6}$$

The loss function in Eqn. 6 can be depicted in the diagram shown in Fig. 1a.

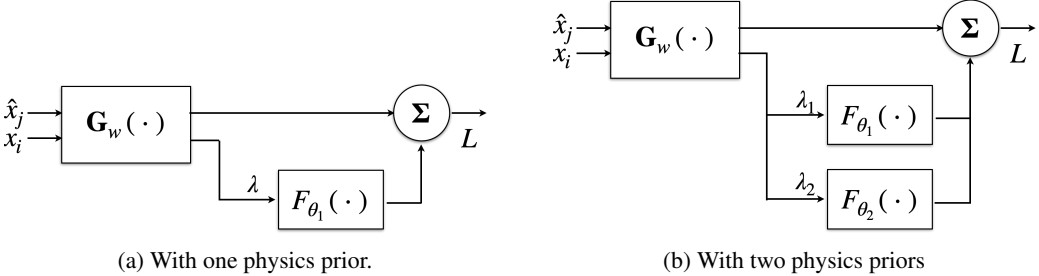

(a) With one physics prior.   (b) With two physics priors

Figure 1: Diagram of model with generalized regularization.

## 2.1 Information Injected by the Generalized Regularizer

Following the SRM inductive principle, the balance between the empirical loss in Eqn. 1 and the structural loss in Eqn. 5 is crucial to ensure good accuracy and generalization performance. To illustrate the essence of the structural loss, we assume the underlying complex system can be described by an "oracle" (but completely unknown to us) model, as shown in Eqn. 7. Note here we promiscuously use $\mathcal{F}_{\tilde{\theta}}(\cdot)$ to represent the oracle model. However in reality it may have no resemblance to the approximate model $\mathcal{F}_\theta$ at all.

$$\mathcal{F}_{\tilde{\theta}}(u)(x) = 0 \quad \forall x \in [0, T] \times D \tag{7}$$

We assume the model trained on the data $\{\hat{x}_j, \hat{y}_j\}$:

$$w^* = \operatorname*{argmin}_w \mathcal{L}_d \tag{8}$$

is the perfect representation of the oracle model $\mathcal{F}_{\tilde{\theta}}$. Hence

$$\mathcal{F}_{\tilde{\theta}}\left(\mathbf{G}_{w^*}(x)\right) = 0 \quad \forall x \in [0, T] \times D \tag{9}$$

To quantify the structural loss shown in Eqn. 5, we have:

$$\mathcal{F}_\theta\left(\mathbf{G}_{w^*}(x)\right) = \mathcal{F}_\theta\left(\mathbf{G}_{w^*}(x)\right) - \mathcal{F}_{\tilde{\theta}}\left(\mathbf{G}_{w^*}(x)\right) \tag{10}$$

$$= \left(\mathcal{F}_\theta(\cdot) - \mathcal{F}_{\tilde{\theta}}(\cdot)\right)\left(\mathbf{G}_{w^*}(x)\right) \quad \forall x \in [0, T] \times D \tag{11}$$

Observe that $\mathcal{F}_\theta(\cdot) - \mathcal{F}_{\tilde{\theta}}(\cdot)$ is the representation of the epistemic uncertainty of our physics prior $\mathcal{F}_\theta$ with respect to the oracle model of the underlying complex system, hence effectively the function space in Eqn. 11 is a projection of the epistemic uncertainty onto the function space of the trained model $\mathbf{G}_w$. Furthermore the regularizer presented in Eqn. 5 is its empirical version, sampled at the collocation points.

When the physics prior has no epistemic uncertainty, theoretically we have $\mathcal{F}_\theta(\cdot) \equiv \mathcal{F}_{\tilde{\theta}}(\cdot)$. Since there still exist aleatoric uncertainties in the observation data, the structural risk term in Eqn. 5 will approach its minimum, but will not automatically become zero. Under this condition, the maximal value of the regularization parameter $\lambda$ will inject most information to the loss function.

---

[1]For notational simplicity, we included the losses from boundary/initial conditions of the priors in $\mathcal{L}_d$.

The immediate consequence of the above observation is that because we usually don't have a concrete metric on the epistemic uncertainty of the physics prior $\mathcal{F}_\theta$, we don't have a good measure on the amount of information our generalized regularizer in Eqn. 5 contains. Hence it is necessary to optimize both the weights and the parameter $\lambda$:

$$w^* = \underset{w,\lambda}{\mathrm{argmin}}\, \mathcal{L} = \underset{w,\lambda}{\mathrm{argmin}}\, \mathcal{L}_d + \lambda \mathcal{L}_p \tag{12}$$

In this study, we perform the optimization by two nested loops:

$$w^* = \underset{\lambda}{\mathrm{argmin}}\, \underset{w}{\mathrm{argmin}}\, \mathcal{L}_d + \lambda \mathcal{L}_p \tag{13}$$

Although the optimization of the model weights $w$ and parameter $\lambda$ outlined in Eqn. 12 and Eqn. 13 appears similar to the optimization of more traditional regularization techniques, such as weight decay in Eqn. 3, we'd like to point out that the generalized regularization place a more strict structural on the solution space of the trained model, hence it is much more cognizant to the behavior of the underlying complex system. This is clearly demonstrated in the experimental results later in the paper.

## 2.2 Regularization with Multiple Physics Priors

An immediate extension following the discussion above is that we can introduce multiple physics priors as generalized regularizers. As the simplest format, these physics priors can be based on the same family of the approximate models, but with different coefficients. The diagram with two physics priors is shown in Fig. 1b. Mathematically the loss function becomes:

$$\mathcal{L} = \mathcal{L}_d + \lambda_1 \mathcal{L}_{p_1} + \lambda_2 \mathcal{L}_{p_2} \tag{14}$$

$$= \underbrace{\frac{1}{N_d} \sum_j^{N_d} (\hat{y}_j - \mathbf{G}_w(\hat{x}_j))^2}_{\mathcal{L}_d} + \underbrace{\frac{\lambda_1}{N_{p_1}} \sum_i^{N_{p_1}} (\mathcal{F}_{\theta_1}(\mathbf{G}_w(x_i))^2}_{\lambda_1 \mathcal{L}_{p_1}} + \underbrace{\frac{\lambda_2}{N_{p_2}} \sum_k^{N_{p_2}} (\mathcal{F}_{\theta_2}(\mathbf{G}_w(x_k))^2}_{\lambda_2 \mathcal{L}_{p_2}} \tag{15}$$

Again we promiscuously denote two physics priors as $\mathcal{F}_{\theta_1}(\cdot)$ and $\mathcal{F}_{\theta_2}(\cdot)$, even though they could be based on completely different families of functions or different families of differential equations. The learning process involves both the regularization parameters and model weights: In this study, we use two nested loops for the optimization:

$$w^* = \underset{\lambda_1,\lambda_2}{\mathrm{argmin}}\, \underset{w}{\mathrm{argmin}}\, \mathcal{L}_d + \lambda_1 \mathcal{L}_{p_2} + \lambda_2 \mathcal{L}_{p_2} \tag{16}$$

Here we make the inductive assumption that the projections of the epistemic uncertainties of the two physics priors should be summed algebraically to provide structural risk minimization. Since the structural risk terms are non-negative, the overall outcome of the risk minimization can be interpreted as a data-driven approach to "select" which physics prior should have a stronger influence on the overall structural risk.

## 2.3 Inclusion of Physics Prior Coefficients

Our generalized regularization method can be further extended to include the coefficients (all or a pre-selected subset) of the physics priors as part of the parameters. More specifically the learning task becomes:

$$w^* = \underset{\lambda,\theta}{\mathrm{argmin}}\, \underset{w}{\mathrm{argmin}}\, \mathcal{L}_d + \lambda \mathcal{L}_p \tag{17}$$

$$= \underset{\lambda,\theta}{\mathrm{argmin}}\, \underset{w}{\mathrm{argmin}} \left[ \frac{1}{N_d} \sum_j^{N_d} (\hat{y}_j - \mathbf{G}_w(\hat{x}_j))^2 + \frac{\lambda}{N_p} \sum_i^{N_p} (\mathcal{F}_\theta(\mathbf{G}_w(x_i))^2 \right] \tag{18}$$

This learning task can be interpreted as simultaneously adjusting the structure of the regularization by optimizing the physics priors represented by $\theta$ and the "strength" of the structural risk by optimizing $\lambda$. The byproduct of the optimization in Eqn. 18, $\theta^*$, from:

$$\theta^* = \underset{\lambda,\theta}{\mathrm{argmin}}\, \underset{w}{\mathrm{argmin}}\, \mathcal{L}_d + \lambda \mathcal{L}_p \tag{19}$$

can be interpreted as the physics prior with the smallest epistemic uncertainty, as the projection to the function space of the trained model.

# 3   Experimental Results

In this section, we present experimental results based on our method. All experiments are written in Pytorch. The training and evaluation are conducted on an Nvidia DGX-2 server with A100 GPUs. More technical details are included in the supplemental materials section.

## 3.1   Implementation in Hamiltonian Neural Networks

In the first example, we forked the public repo of the Hamiltonian Neural Network(HNN)[7] and implemented generalized regularizer. The elegant utilization of Hamiltonian mechanics in HNN makes it straight-forward to implement generalized regularizer based on physics priors. For each case, the physics prior is parameterized by another Hamiltonian $\mathcal{H}_\theta$, where $\theta$ represents the parameters of physics models such as mass, length of the pendulum. We introduce the generalized regularization by:

$$\mathcal{L}_{reg} = \lambda \cdot \left\| \left( \frac{\partial \mathcal{H}_w}{\partial \mathbf{p}} - \frac{\partial \mathcal{H}_\theta}{\partial \mathbf{p}} \right) + \left( \frac{\partial \mathcal{H}_w}{\partial \mathbf{q}} - \frac{\partial \mathcal{H}_\theta}{\partial \mathbf{q}} \right) \right\|_2 \tag{20}$$

while the original HNN loss is computed by:

$$\mathcal{L}_{HNN} = \left\| \frac{\partial \mathcal{H}_w}{\partial \mathbf{p}} - \frac{\partial \mathbf{q}}{\partial t} \right\|_2 + \left\| -\frac{\partial \mathcal{H}_w}{\partial \mathbf{q}} - \frac{\mathbf{q}}{\partial t} \right\|_2 \tag{21}$$

where $w$ denotes the trainable weights of the HNN model.

Results of three cases are presented in Tab. 1: mass-spring, ideal pendulum and real pendulum. Improvements are illustrated in Fig. 2. In the last example, the training data are collected from measurements of physical pendulum, which is subject to sensor noise, as well as epistemic uncertainties such as frictions[22]. In all three cases, the generalized regularization demonstrated clear improvements in both baseline model and HNN model. We want to point out that for ideal pendulum, generalized regularization further improves the energy by $10\times$, in addition to $2\times$ improvement of HNN. In the case of real pendulum, in which a precise physics model is unknown, HNN demonstrated an impressive $30\times$ improvement in terms of energy (from $376.9$ to $11.2$), while the introduction of physics prior can further improve the energy metric to $9.5$, an additional $15\%$ improvement.

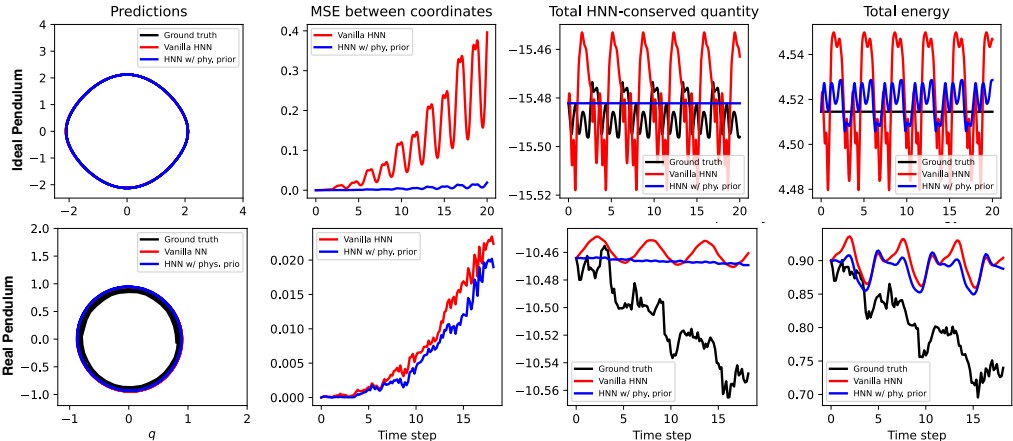

Figure 2:  Performances of Hamiltonian NN with and without physics regularizers. Generalized regularization implementation is based on the open-source repo of Greydanus et. al. "Hamiltonian Neural Networks".

| | Test Loss | | Energy | |
| --- | --- | --- | --- | --- |
| Task | Baseline | HNN | Baseline | HNN |
| Mass spring (original) | $36.73 \pm 1.86$ | $35.91 \pm 1.83$ | $\mathbf{147.01 \pm 19.30}$ | $0.376 \pm 0.077$ |
| Mass spring (w/ reg.) | $\mathbf{36.55 \pm 1.85}$ | $\mathbf{35.90 \pm 1.83}$ | $167.91 \pm 20.50$ | $\mathbf{0.364 \pm 0.084}$ |
| Ideal Pendulum (original) | $35.32 \pm 1.80$ | $35.59 \pm 1.82$ | $41.83 \pm 9.75$ | $24.85 \pm 5.42$ |
| Ideal Pendulum (w/ reg.) | $\mathbf{35.11 \pm 1.78}$ | $\mathbf{34.56 \pm 1.74}$ | $\mathbf{34.56 \pm 8.55}$ | $\mathbf{2.29 \pm 0.47}$ |
| Real Pendulum (original) | $1.50 \pm 0.23$ | $5.80 \pm 0.60$ | $376.89 \pm 72.50$ | $11.22 \pm 3.87$ |
| Real Pendulum (w/ reg.) | $\mathbf{1.39 \pm 0.20}$ | $\mathbf{5.79 \pm 0.62}$ | $\mathbf{371.50 \pm 74.30}$ | $\mathbf{9.46 \pm 3.70}$ |

Table 1: Quantitative results of three tasks in Table 1 in Geydanus et. al. "Hamiltonian Neural Networks". All values are multiplied by $10^3$. Implementation is based on the Github repo by Geydanus, although some values of the original models are slightly different from values reported in the original paper. **Bold** entries indicate the best results.

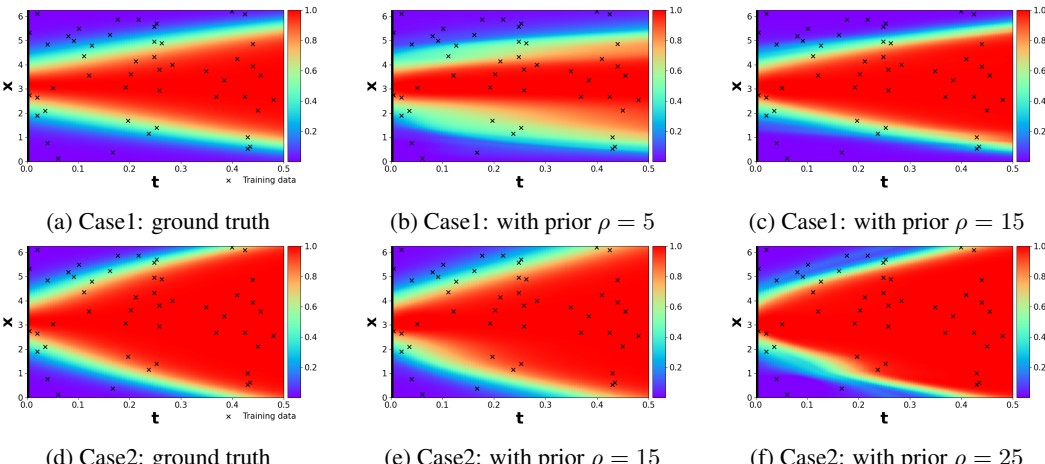

(a) Case1: ground truth     (b) Case1: with prior $\rho = 5$     (c) Case1: with prior $\rho = 15$

(d) Case2: ground truth     (e) Case2: with prior $\rho = 15$     (f) Case2: with prior $\rho = 25$

Figure 3: Two cases of 1D reaction equation, including the ground truth and models with different physics priors. The crosses indicate the collocation points of training data.

## 3.2 1D Reaction Equation

We study the one dimensional reaction equation that is commonly used to model chemical reactions. It's the simplification of the reaction-diffusion equations and is given by,

$$\frac{\partial u}{\partial t} - \rho u(1 - u) = 0 \,, \tag{22}$$

with the associated initial and (periodic) boundary conditions $u(x, 0) = g(x)$, $x \in D$ and $u(0, t) = u(2\pi, t)$, $t \in (0, T]$ respectively, with $g(x) = \exp\left(-\frac{(x-\pi)^2}{2(\pi/4)^2}\right)$, with $\rho$ being the reaction coefficient. We generate two datasets by using two oracle reaction equations, shown in Fig. 3a and 3d respectively. Fig. 3 also shows results of learned models with different physics priors, all are different from the oracle models.

As a comparison, we also train the baseline models for the two reaction equation cases from observation data only, with and without using weight decay as regularization. The results are shown in Fig. 4. Quantitative MSE from testing are tabulated in Tab. 2. Clearly our method outperforms weigh decay in all but one case.

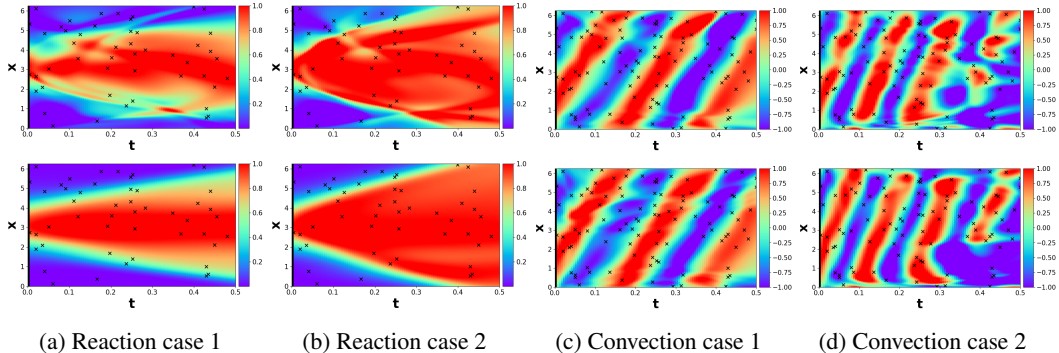

| (a) Reaction case 1 | (b) Reaction case 2 | (c) Convection case 1 | (d) Convection case 2 |

Figure 4: Baseline performance without regularization (top row) and with weight decay (bottom row). Left two columns represent reaction equation cases and the right two columns are convection cases.

Table 2: Testing MSE for 1D Reaction Equation

|  | prior used | baseline w/o reg. | baseline w/ weight decay | $\lambda = \lambda_{opt}$ |  |
|---|---|---|---|---|---|
| case 1 | $\rho = 5$ | $1.87 \times 10^{-2}$ | $\mathbf{3.61 \times 10^{-3}}$ | $6.67 \times 10^{-3}$ | $\lambda_{opt} = 8.97 \times 10^{-1}$ |
|  | $\rho = 15$ |  | $3.61 \times 10^{-3}$ | $\mathbf{2.31 \times 10^{-3}}$ | $\lambda_{opt} = 4.51 \times 10^{-2}$ |
| case 2 | $\rho = 15$ | $1.88 \times 10^{-2}$ | $3.98 \times 10^{-3}$ | $\mathbf{1.66 \times 10^{-3}}$ | $\lambda_{opt} = 5.61 \times 10^{-2}$ |
|  | $\rho = 25$ |  | $3.98 \times 10^{-3}$ | $\mathbf{2.00 \times 10^{-3}}$ | $\lambda_{opt} = 1.13 \times 10^{-2}$ |

### 3.3  1D Convection Equation

One-dimensional convection refers to the process of transport or flow of a fluid or a scalar quantity in a 1D domain. It is commonly described by the following hyperbolic PDE:

$$\frac{\partial u}{\partial t} + \beta \frac{\partial u}{\partial x} = 0, \; x \in D, t \in [0, T], \tag{23}$$

with the initial condition $u(x, 0) = \sin x$ and periodic boundary condition $u(0, t) = u(2\pi, t)$.

We repeat the experiment. The performance heatmaps are shown in Fig. 5 and quantitative testing MSE tabulated in Tab. 3. Compared to the baseline models without regularization and with weight decay, as shown in Fig. 4, our method demonstrated substantial improvements.

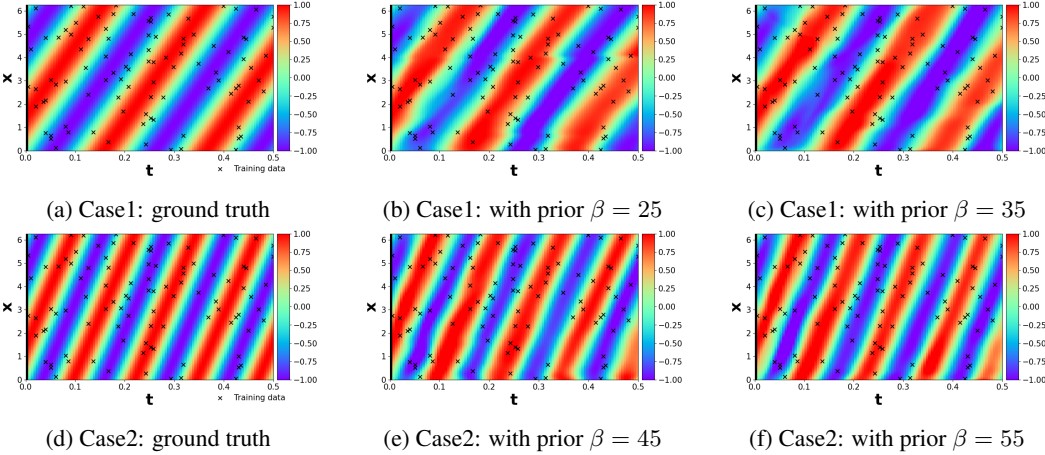

| (a) Case1: ground truth | (b) Case1: with prior $\beta = 25$ | (c) Case1: with prior $\beta = 35$ |
| (d) Case2: ground truth | (e) Case2: with prior $\beta = 45$ | (f) Case2: with prior $\beta = 55$ |

Figure 5: Two cases of 1D convection equation, including the ground truth and models with different physics priors.

Table 3: Testing MSE for 1D Convection Equation and associated optimal $\lambda$

|  | prior used | baseline w/o reg. | baseline w/ weight decay | $\lambda = \lambda_{opt}$ |  |
|---|---|---|---|---|---|
| case 1 | $\beta = 25$ | $7.85 \times 10^{-2}$ | $4.33 \times 10^{-2}$ | $\mathbf{1.29 \times 10^{-2}}$ | $\lambda_{opt} = 8.42 \times 10^{-2}$ |
|  | $\beta = 35$ |  |  | $\mathbf{1.16 \times 10^{-2}}$ | $\lambda_{opt} = 1.16 \times 10^{-2}$ |
| case 2 | $\beta = 45$ | $6.37 \times 10^{-1}$ | $5.11 \times 10^{-1}$ | $\mathbf{1.57 \times 10^{-2}}$ | $\lambda_{opt} = 2.42 \times 10^{-2}$ |
|  | $\beta = 55$ |  |  | $\mathbf{1.16 \times 10^{-2}}$ | $\lambda_{opt} = 3.58 \times 10^{-2}$ |

Finally we use case 1 of the convection equation experiment for demonstration of multiple physics priors and optimization of physics prior coefficients, shown in Fig. 6. In the first experiment, two physics priors are specified with $\beta = 25$ and $\beta = 35$. The optimal regularization parameters are $\lambda_1^* = 3.72 \times 10^{-7}$ and $\lambda_2^* = 3.47 \times 10^{-3}$ with testing MSE of $9.53 \times 10^{-3}$. In the second experiment, the final physics prior coefficient is $\beta^* = 29.67$. With an optimal $\lambda^* = 1.25 \times 10^{-1}$, it achieves the testing MSE of $5.04 \times 10^{-3}$.

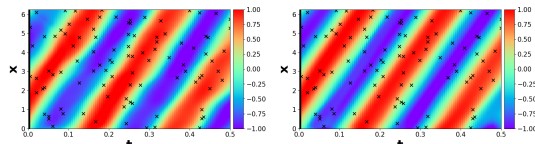

(a) With two physics priors.    (b) With physics coef.

Figure 6: Performance heatmap of case 1 of convection equation

## 4 Conclusion and Future Work

In this paper we propose a principled method to incorporate the prior knowledge of an underlying complex system, in the form of approximate physics models, into the data-driven deep grey-box modeling. By structuring the imprecise physics models, or physics priors, as generalized regularizers, we apply Vapnik's structural risk minimization (SRM) inductive principle to balance the model accuracy and model complexity. Our analysis indicates that the information in the physics priors is bounded by the uncertainty, especially the epistemic uncertainty, of the physics priors. Experimental results have shown that our method is highly effective in improving the test accuracy. For future work, we plan to investigate the theoretical and practical implications when multiple physics priors are included in the regularization.

## Acknowledgement

This research was supported by the U.S. Department of Energy, through the Office of Advanced Scientific Computing Research's "Data-Driven Decision Control for Complex Systems (DnC2S)" project, FWP ERKJ368. This manuscript has been authored in part by UT-Battelle, LLC under Contract No. DE-AC05-00OR22725 with the U.S. Department of Energy. The United States Government retains and the publisher, by accepting the article for publication, acknowledges that the United States Government retains a non-exclusive, paid-up, irrevocable, world-wide license to publish or reproduce the published form of this manuscript, or allow others to do so, for United States Government purposes. The Department of Energy will provide public access to the results of federally sponsored research in accordance with the DOE Public Access Plan (https://energy.gov/downloads/doe-public-access-plan).

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

# Deep Learning with Physics Priors as Generalized Regularizers

## Supplemental Material

**Frank Liu**
School of Data Science
Old Dominion University
Norfolk, VA 23508
United States

**Agniva Chowdhury**
Computer Science and Math Division
Oak Ridge National Lab
Oak Ridge, TN 37831
United States

## S1  Inclusion of Initial and Boundary Conditions in the Loss Function

When the initial and/or boundary conditions are available for the underlying dynamic system, they should be included in the loss function, namely $\mathcal{L}_d$ in Eqn. (6) of the main text. More specifically the component due to the initial conditions is:

$$\mathcal{L}_I = \frac{1}{N_I} \sum_j^{N_I} \left( \hat{y}_j^I - \mathbf{G}_w([\hat{x}_j^I, 0]) \right)^2 \tag{1}$$

and the component due to the boundary condition is:

$$\mathcal{L}_b = \frac{1}{N_b} \sum_j^{N_b} \left( \mathbf{G}_w([0, \hat{t}_j^b]) - \mathbf{G}_w([2\pi, \hat{t}_j^b]) \right)^2 \tag{2}$$

Here we explicitly separate temporal and spatial component in the input $\hat{x}_j$. These terms should be combined with the $\mathcal{L}_d$ term in the overall loss function.

## S2  Experiment Setup

All experiments are conducted on an Nvidia A100 GPU with 40GB of memory. The DL framework used is PyTorch 1.10.1. The optimization is conducted using Bayesian Optimization routine in Scikit-Optimize version 0.9.0. In both reaction and convection cases, the DNN model is an MLP with 5 hidden layers, each with a width of 512, and `tanh` as the activation function. `Adam` is used as the optimizer, with initial learning rate set to `LR` $= 2 \times 10^{-4}$.

## S3  Details and Additional Results on 1D Reaction Equation

### S3.1  Details of Experiment Setup

The reaction equation is specified as:

$$\frac{\partial u}{\partial t} - \rho u(1 - u) = 0 \tag{3}$$

We generate data points using "oracle" models which are also reaction equations. For each case, we consider a mesh which consists of 100 time points between $T = [0, 0.5]$, with 256 spatial points at each time point. This results in a total of 25,600 grid points. All 256 spatial points for $T = 0$ are included in the loss function term related to the initial condition in Eqn. 1. Therefore, we have $([\hat{x}_j^I, 0], \hat{y}_j^I)$, where $\hat{y}_j^I = u(\hat{x}_j^I, 0)$ for $j = 1, 2, \ldots, N_I$, and $N_I = 256$. Similarly, we compute

the boundary loss for the periodic boundary condition using Eqn. (2) for the boundary time points $[0, \hat{t}_1^b], [0, \hat{t}_2^b], \ldots, [0, \hat{t}_{N_b}^b]$, with $N_b = 100$. Additional 50 data points are randomly chosen to be included in the loss function $\mathcal{L}_d$, with $\mathcal{N}(0, 0.1)$ noise added to the them. Two "oracle" models are considered:

- Case 1: $\rho = 10$
- Case 2: $\rho = 20$

We want to emphasize that the oracle models are never used as the physics prior.

Finally we randomly choose 100 collocation points in the support to compute generalized regularizers based on the physics priors. To provide quantitative measure on the accuracy of the learned model, we use all remaining points to compute mean square error (MSE) of the trained models.

## S3.2  Additional Experiments

Under this setting, we vary the regularization parameter $\lambda$ using predefined values: $\lambda = \{10^{-5}, 10^{-3}, 10^{-2}, 10^{-1}\}$. Only one physics prior is used, chosen from the following: $\rho = \{5, 15\}$ for Case 1 and $\rho = \{15, 25\}$ for Case 2. We include weight decay as one of the baseline results. To provide a fair comparison, we use four decay parameters $\{10^{-3}, 10^{-4}, 10^{-6}, 10^{-8}\}$, and choose the best outcome.

The results are shown in Fig. S1, Fig. S2, Fig. S3 and Fig. S4 respectively. In each case, besides the two baseline results (with and without weight decay), we also included the results from the optimal $\lambda$ from optimization. Quantitative comparisons are summarized in Tab. S1 and Tab. S2.

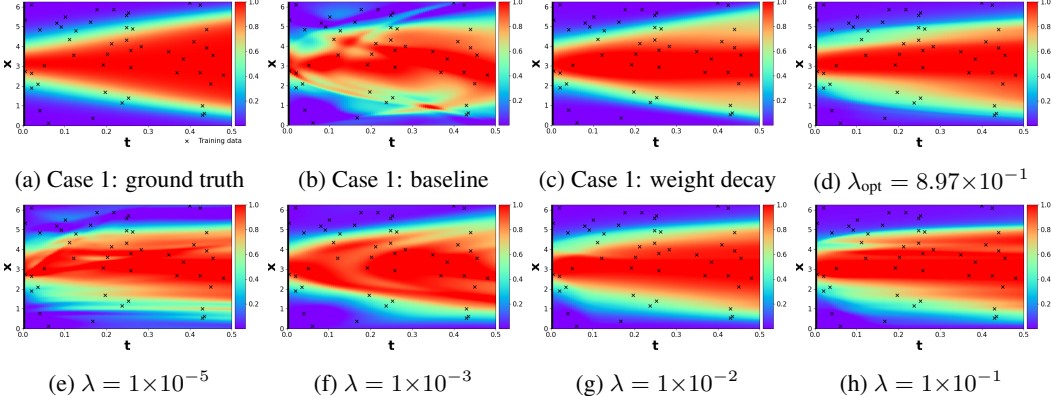

(a) Case 1: ground truth  (b) Case 1: baseline  (c) Case 1: weight decay  (d) $\lambda_{\text{opt}} = 8.97 \times 10^{-1}$

(e) $\lambda = 1 \times 10^{-5}$  (f) $\lambda = 1 \times 10^{-3}$  (g) $\lambda = 1 \times 10^{-2}$  (h) $\lambda = 1 \times 10^{-1}$

Figure S1: Heatmap of predicted solutions for Case 1 of 1D reaction, with $\rho = 5$ as the physics prior. Top row: (a) Exact solution, (b) Baseline solution (no physics prior, no weight decay), (c) Baseline solution with weight decay, and (d) Solution obtained after tuning $\lambda$ using BO with physics prior $\rho = 5$. Bottom row: (e)-(h) Solutions for each pre-specified values of $\lambda$ with physics prior $\rho = 5$. Note that the quality of the output can be significantly enhanced when employing BO to tune the values of $\lambda$ as in (d). See Tables S1 and S2 for the corresponding test MSEs.

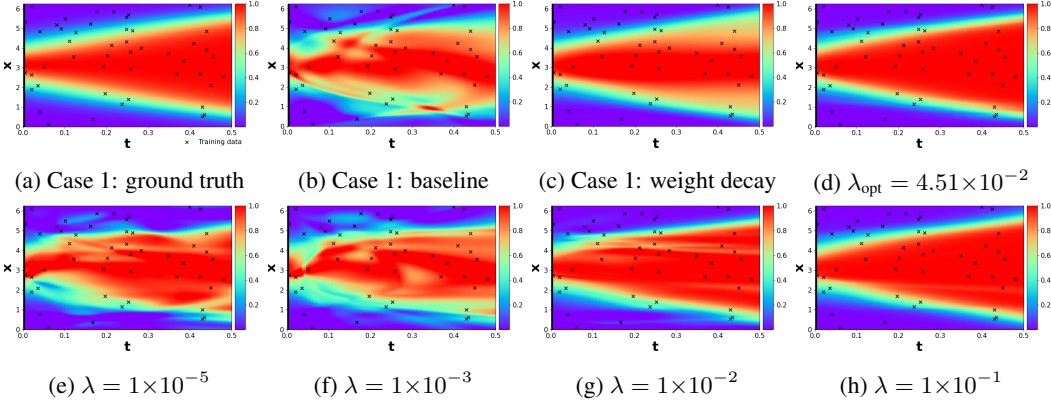

(a) Case 1: ground truth     (b) Case 1: baseline     (c) Case 1: weight decay     (d) $\lambda_{\text{opt}} = 4.51 \times 10^{-2}$

(e) $\lambda = 1 \times 10^{-5}$     (f) $\lambda = 1 \times 10^{-3}$     (g) $\lambda = 1 \times 10^{-2}$     (h) $\lambda = 1 \times 10^{-1}$

Figure S2: Heatmap of predicted solutions for case 1 of 1D reaction with $\rho = 15$ as the physics prior. Top row: (a) Exact solution, (b) Baseline solution (no physics prior, no weight decay), (c) Baseline with eight decay, and (d) Solution obtained after tuning $\lambda$ using BO with physics prior $\rho = 15$. Bottom row: (e)-(h) Solutions for each pre-specified values of $\lambda$ with physics prior $\rho = 15$. See Tables S1 and S2 for quantitative comparisons.

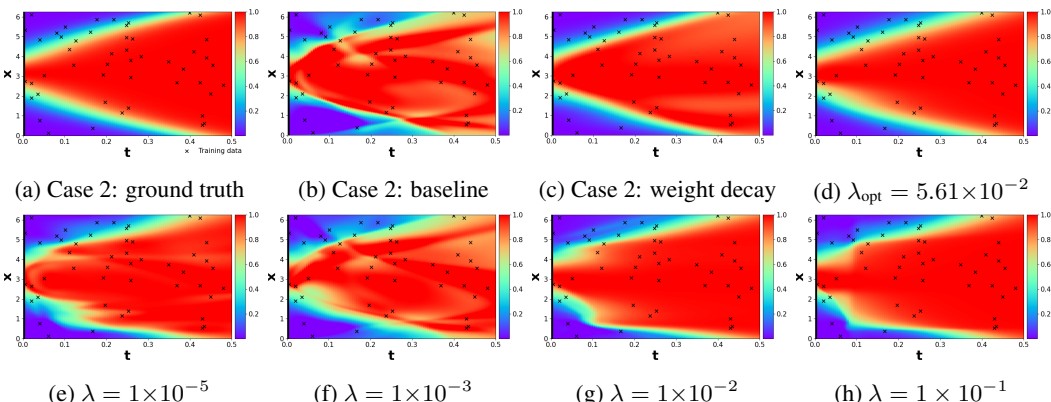

(a) Case 2: ground truth     (b) Case 2: baseline     (c) Case 2: weight decay     (d) $\lambda_{\text{opt}} = 5.61 \times 10^{-2}$

(e) $\lambda = 1 \times 10^{-5}$     (f) $\lambda = 1 \times 10^{-3}$     (g) $\lambda = 1 \times 10^{-2}$     (h) $\lambda = 1 \times 10^{-1}$

Figure S3: Heatmap of predicted solutions for Case 2 of 1D reaction with $\rho = 15$ as the physics prior. Top row: (a) Exact solution, (b) Baseline solution (no physics prior, no weight decay), (c) Baseline with weight decay, and (d) Solution obtained after tuning $\lambda$ using BO with physics prior $\rho = 15$. Bottom row: (e)-(h) Solutions for each pre-specified values of $\lambda$ with physics prior $\rho = 15$.

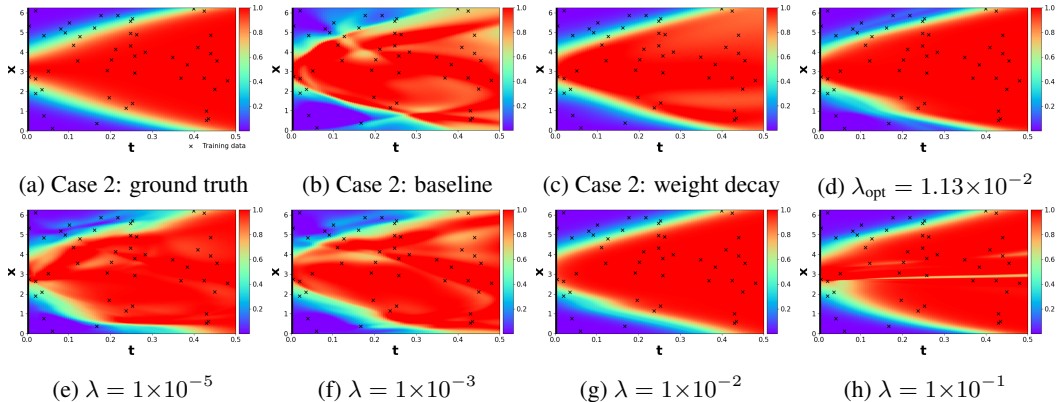

(a) Case 2: ground truth    (b) Case 2: baseline    (c) Case 2: weight decay    (d) $\lambda_{\text{opt}} = 1.13{\times}10^{-2}$

(e) $\lambda = 1{\times}10^{-5}$    (f) $\lambda = 1{\times}10^{-3}$    (g) $\lambda = 1{\times}10^{-2}$    (h) $\lambda = 1{\times}10^{-1}$

Figure S4: Heatmap of predicted solutions for case 2 of 1D reaction with $\rho = 25$ as the physics prior. Top row: (a) Exact solution, (b) Baseline solution (no physics prior, no weight decay), (c) Baseline with weight decay, and (d) Solution obtained after tuning $\lambda$ using BO with physics prior $\rho = 25$. Bottom row: (e)-(h) Solutions for each pre-specified values of $\lambda$ with physics prior $\rho = 25$.

Table S1: Testing MSE for 1D Reaction Equation. The last column indicates the optimal $\lambda$ values from optimization.

|  | prior used | baseline w/o reg. | baseline w/ weight decay | $\lambda = \lambda_{opt}$ |  |
|---|---|---|---|---|---|
| case 1 | $\rho = 5$ | $1.87{\times}10^{-2}$ | $\mathbf{3.61{\times}10^{-3}}$ | $6.67{\times}10^{-3}$ | $\lambda_{opt} = 8.97{\times}10^{-1}$ |
|  | $\rho = 15$ |  | $3.61{\times}10^{-3}$ | $\mathbf{2.31{\times}10^{-3}}$ | $\lambda_{opt} = 4.51{\times}10^{-2}$ |
| case 2 | $\rho = 15$ | $1.88{\times}10^{-2}$ | $3.98{\times}10^{-3}$ | $\mathbf{1.66{\times}10^{-3}}$ | $\lambda_{opt} = 5.61{\times}10^{-2}$ |
|  | $\rho = 25$ |  | $3.98{\times}10^{-3}$ | $\mathbf{2.00{\times}10^{-3}}$ | $\lambda_{opt} = 1.13{\times}10^{-2}$ |

Table S2: 1D reaction: Test MSE for prespecified values of $\lambda$ and for the optimal $\lambda$.

| Oracle | Prior | Baseline | Weight decay | $\lambda = 1 \times 10^{-5}$ | $\lambda = 1 \times 10^{-3}$ | $\lambda = 1 \times 10^{-2}$ | $\lambda = 1 \times 10^{-1}$ | $\lambda = \lambda_{\text{opt}}$ |
|---|---|---|---|---|---|---|---|---|
| case 1 | $\rho = 5$ | $1.87 \times 10^{-2}$ | $3.61 \times 10^{-3}$ | $7.68 \times 10^{-3}$ | $1.64 \times 10^{-2}$ | $6.39 \times 10^{-3}$ | $1.17 \times 10^{-2}$ | $6.67 \times 10^{-3}$ |
|  | $\rho = 15$ |  |  | $1.63 \times 10^{-2}$ | $1.66 \times 10^{-2}$ | $6.58 \times 10^{-3}$ | $3.74 \times 10^{-3}$ | $\mathbf{2.31 \times 10^{-3}}$ |
| case 2 | $\rho = 15$ | $1.88 \times 10^{-2}$ | $3.98 \times 10^{-3}$ | $8.83 \times 10^{-3}$ | $3.39 \times 10^{-3}$ | $5.12 \times 10^{-3}$ | $1.83 \times 10^{-3}$ | $\mathbf{1.66 \times 10^{-3}}$ |
|  | $\rho = 25$ |  |  | $1.45 \times 10^{-2}$ | $1.17 \times 10^{-2}$ | $2.43 \times 10^{-3}$ | $4.44 \times 10^{-3}$ | $\mathbf{2.00 \times 10^{-3}}$ |

### S3.3 Experiments with Multiple Physics Priors and Co-optimization of Physics Coefficients

In addition, we use Case 2 of the reaction equation experiment for demonstration of multiple physics priors and co-optimization of physics prior coefficients, as shown in Fig. S5. In the first experiment, two physics priors are used, which are specified with $\rho = 15$ and $\rho = 25$. The optimal regularization parameters are $\lambda_1^* = 1.10{\times}10^{-7}$ and $\lambda_2^* = 7.65{\times}10^{-3}$ with the testing MSE of $1.03{\times}10^{-3}$. Observe that the optimized $\lambda$ of the second physics prior $\rho = 25$ is much larger than the parameter of the first prior. Also note the testing MSE is smaller than the individual cases when only one physics prior is used, as shown in Tab. S1.

In the second experiment, the final physics prior coefficient after optimization is $\rho^* = 19.95$. With an optimal $\lambda^* = 2.53{\times}10^{-2}$, it achieves the testing MSE of $1.09{\times}10^{-3}$. See Table S9 for details.

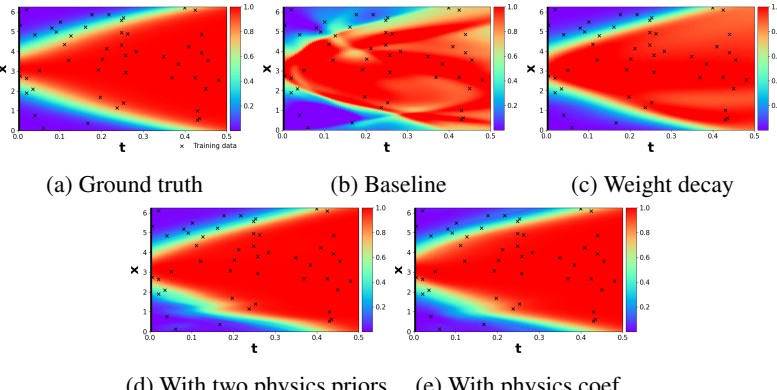

(a) Ground truth      (b) Baseline      (c) Weight decay

(d) With two physics priors      (e) With physics coef.

Figure S5: Performance heatmap of case 2 of reaction equation. The first row represents the ground truth and performance of baseline models (with and without weight decay). In the second row, (d) shows results with two physics priors, while (e) with co-optimization of the physics coefficient $\rho$.

Table S3: Testing MSE for Case2 of 1D Reaction Equation: Multiple priors and Physics coefficients. The last column indicates the optimal $\lambda$ values, and the physics coefficients, respectively.

|  | baseline w/o reg. | baseline w/ weight decay | $\lambda = \lambda_{opt}$ |  |
|---|---|---|---|---|
| Multiple priors | $1.88{\times}10^{-2}$ | $3.98{\times}10^{-3}$ | $1.03 \times 10^{-3}$ | $\lambda_{opt_1} = 1.10{\times}10^{-7}$ (Prior 1: $\rho = 15$) $\lambda_{opt_2} = 7.65{\times}10^{-3}$ (Prior 2: $\rho = 25$) |
| Optimizing phys coeffs. | $1.88{\times}10^{-2}$ | $3.98{\times}10^{-3}$ | $1.09 \times 10^{-3}$ | $\lambda_{opt} = 2.53{\times}10^{-2}$ $\rho_{opt} = 19.95$ |

# S4 Additional Experimental Results on 1D Convection Equation

The 1D convection equation is specified by:

$$\frac{\partial u}{\partial t} + \beta \frac{\partial u}{\partial x} = 0, \ x \in D, t \in [0, T], \tag{4}$$

The experiment setup is similar to that of the reaction case. We generated the data based on two "oracle" models:

- Case 1: $\beta = 30$
- Case 2: $\beta = 50$

We repeat the same set of experiments as in the 1D reaction case. For Case 1, we choose the physics prior from: $\beta = \{25, 35\}$. For Case 2, we choose physics prior from $\beta = \{45, 55\}$.

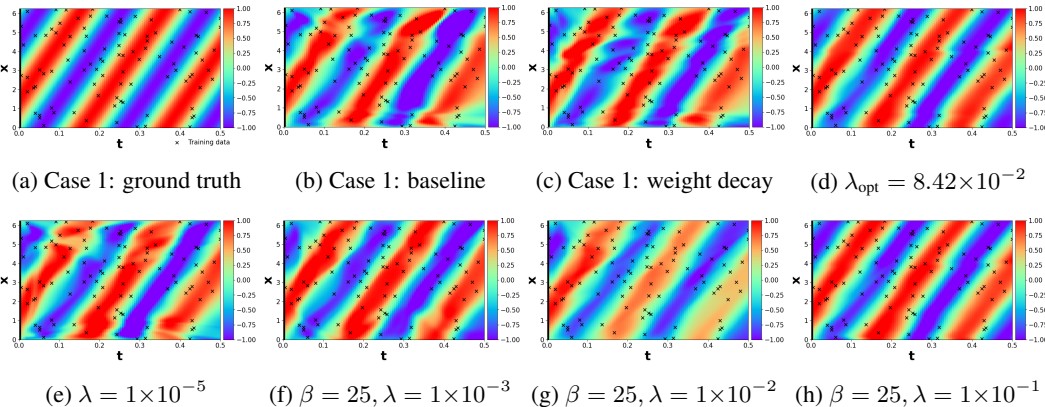

(a) Case 1: ground truth    (b) Case 1: baseline    (c) Case 1: weight decay    (d) $\lambda_{\text{opt}} = 8.42 \times 10^{-2}$

(e) $\lambda = 1 \times 10^{-5}$    (f) $\beta = 25, \lambda = 1 \times 10^{-3}$    (g) $\beta = 25, \lambda = 1 \times 10^{-2}$    (h) $\beta = 25, \lambda = 1 \times 10^{-1}$

Figure S6: Heatmap of predicted solutions for case 1 of 1D convection with $\beta = 25$ as the physics prior. Top row: (a) Exact solution, (b) Baseline solution (no physics prior), (c) Weight decay, and (d) Solution obtained after tuning $\lambda$ using BO with physics prior $\beta = 25$. Bottom row: (e)-(h) Solutions for each pre-specified values of $\lambda$ with physics prior $\beta = 25$. See Tables S4 and S5 for quantitative results.

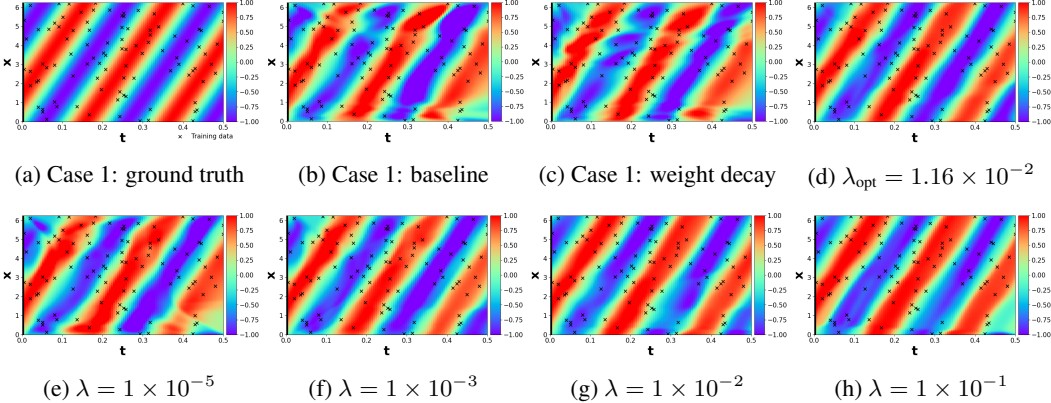

(a) Case 1: ground truth    (b) Case 1: baseline    (c) Case 1: weight decay    (d) $\lambda_{\text{opt}} = 1.16 \times 10^{-2}$

(e) $\lambda = 1 \times 10^{-5}$    (f) $\lambda = 1 \times 10^{-3}$    (g) $\lambda = 1 \times 10^{-2}$    (h) $\lambda = 1 \times 10^{-1}$

Figure S7: Heatmap of predicted solutions for case 1 of 1D convection with $\beta = 35$ as the physics prior. Top row: (a) Exact solution, (b) Baseline solution (no physics prior), (c) Weight decay, and (d) Solution obtained after tuning $\lambda$ using BO with physics prior $\beta = 35$. Bottom row: (e)-(h) Solutions for each pre-specified values of $\lambda$ with physics prior $\beta = 35$.

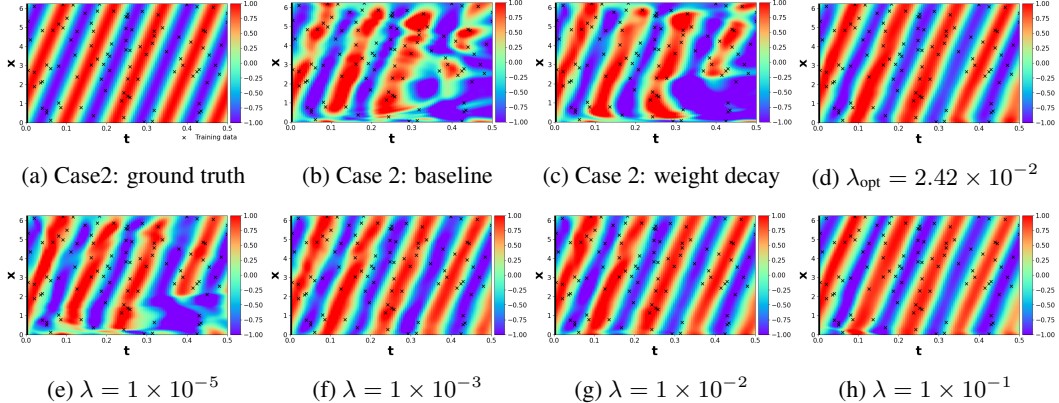

(a) Case2: ground truth     (b) Case 2: baseline     (c) Case 2: weight decay     (d) $\lambda_{\text{opt}} = 2.42 \times 10^{-2}$

(e) $\lambda = 1 \times 10^{-5}$     (f) $\lambda = 1 \times 10^{-3}$     (g) $\lambda = 1 \times 10^{-2}$     (h) $\lambda = 1 \times 10^{-1}$

Figure S8: Heatmap of predicted solutions for case 2 of 1D convection with $\beta = 45$ as the physics prior. Top row: (a) Exact solution, (b) Baseline solution (no physics prior), (c) Weight decay, and (d) Solution obtained after tuning $\lambda$ using BO with physics prior $\beta = 45$. Bottom row: (e)-(h) Solutions for each pre-specified values of $\lambda$ with physics prior $\beta = 45$.

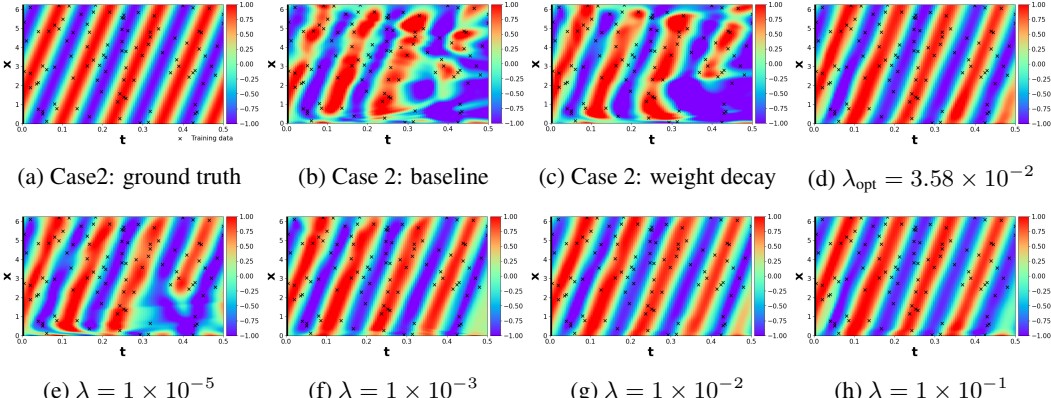

(a) Case2: ground truth     (b) Case 2: baseline     (c) Case 2: weight decay     (d) $\lambda_{\text{opt}} = 3.58 \times 10^{-2}$

(e) $\lambda = 1 \times 10^{-5}$     (f) $\lambda = 1 \times 10^{-3}$     (g) $\lambda = 1 \times 10^{-2}$     (h) $\lambda = 1 \times 10^{-1}$

Figure S9: Heatmap of predicted solutions for case 2 of 1D convection with $\beta = 55$ as the physics prior. Top row: (a) Exact solution, (b) Baseline solution (no physics prior), (c) Weight decay, and (d) Solution obtained after tuning $\lambda$ using BO with physics prior $\beta = 55$. Bottom row: (e)-(h) Solutions for each pre-specified values of $\lambda$ with physics prior $\beta = 55$.

Table S4: Testing MSE for 1D Convection Equation and associated optimal $\lambda$

|  | prior used | baseline w/o reg. | baseline w/ weight decay | $\lambda = \lambda_{opt}$ |  |
|---|---|---|---|---|---|
| case 1 | $\beta = 25$ | $7.85 \times 10^{-2}$ | $4.33 \times 10^{-2}$ | $\mathbf{1.29 \times 10^{-2}}$ | $\lambda_{opt} = 8.42 \times 10^{-2}$ |
|  | $\beta = 35$ |  |  | $\mathbf{1.16 \times 10^{-2}}$ | $\lambda_{opt} = 1.16 \times 10^{-2}$ |
| case 2 | $\beta = 45$ | $6.37 \times 10^{-1}$ | $5.11 \times 10^{-1}$ | $\mathbf{1.57 \times 10^{-2}}$ | $\lambda_{opt} = 2.42 \times 10^{-2}$ |
|  | $\beta = 55$ |  |  | $\mathbf{1.16 \times 10^{-2}}$ | $\lambda_{opt} = 3.58 \times 10^{-2}$ |

Table S5: 1D convection: Test MSEs for prespecified values of $\lambda$.

| Oracle | Prior | Baseline | Weight decay | $\lambda = 1 \times 10^{-5}$ | $\lambda = 1 \times 10^{-3}$ | $\lambda = 1 \times 10^{-2}$ | $\lambda = 1 \times 10^{-1}$ | $\lambda = \lambda_{\text{opt}}$ |
|---|---|---|---|---|---|---|---|---|
| Case 1 | $\beta = 25$ | $7.85 \times 10^{-2}$ | $4.33 \times 10^{-2}$ | $4.50 \times 10^{-2}$ | $2.17 \times 10^{-2}$ | $5.88 \times 10^{-2}$ | $2.17 \times 10^{-2}$ | $\mathbf{1.29 \times 10^{-2}}$ |
| | $\beta = 35$ | | | $5.15 \times 10^{-2}$ | $2.16 \times 10^{-2}$ | $1.65 \times 10^{-2}$ | $2.47 \times 10^{-2}$ | $\mathbf{1.16 \times 10^{-2}}$ |
| Case 2 | $\tilde{\beta} = 45$ | $6.37 \times 10^{-1}$ | $5.11 \times 10^{-1}$ | $2.35 \times 10^{-1}$ | $2.12 \times 10^{-2}$ | $1.94 \times 10^{-2}$ | $1.99 \times 10^{-2}$ | $\mathbf{1.57 \times 10^{-2}}$ |
| | $\tilde{\beta} = 55$ | | | $1.38 \times 10^{-1}$ | $2.44 \times 10^{-2}$ | $2.00 \times 10^{-2}$ | $3.34 \times 10^{-2}$ | $\mathbf{1.16 \times 10^{-2}}$ |

**1D Convection: Multiple Physics Priors and Co-optimization of Physics Coefficients**

In addition, we use case 1 of the convection equation experiment for demonstration of multiple physics priors and optimization of physics prior coefficients, shown in Fig. S10. In the first experiment, two physics priors are specified with $\beta = 25$ and $\beta = 35$. The optimal regularization parameters are $\lambda_1^* = 3.72 \times 10^{-7}$ and $\lambda_2^* = 3.47 \times 10^{-3}$ with testing MSE of $9.53 \times 10^{-3}$. In the second experiment, the final physics prior coefficient is $\beta^* = 29.67$. With an optimal $\lambda^* = 1.25 \times 10^{-1}$, it achieves the testing MSE of $5.04 \times 10^{-3}$. See Table S6 for details.

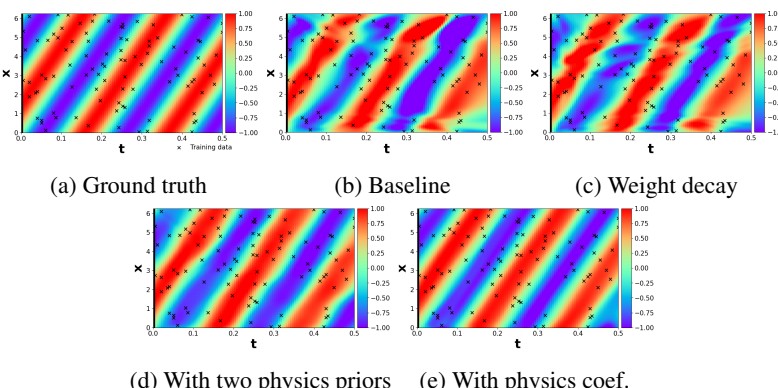

(a) Ground truth       (b) Baseline      (c) Weight decay

(d) With two physics priors      (e) With physics coef.

Figure S10: Performance heatmap of case 1 of convection equation. The first row shows the ground truth and performance of baseline models (with and without weight decay). In the last row, (d) shows the results with two physics priors, while (e) shows results by co-optimizing physics coefficient $\beta$, along with $\lambda$.

Table S6: Testing MSE for Case 1 of 1D Convection Equation: Multiple priors and Physics coefficients. The last column indicates the optimal $\lambda$ values, and the physics coefficients, respectively.

| | baseline w/o reg. | baseline w/ weight decay | $\lambda = \lambda_{opt}$ | |
|---|---|---|---|---|
| Multiple priors | $7.85 \times 10^{-2}$ | $4.33 \times 10^{-2}$ | $9.53 \times 10^{-3}$ | $\lambda_{opt_1} = 3.72 \times 10^{-7}$ (Prior 1: $\beta = 25$) $\lambda_{opt_2} = 3.47 \times 10^{-3}$ (Prior 2: $\beta = 35$) |
| Optimizing phys coeffs. | $7.85 \times 10^{-2}$ | $4.33 \times 10^{-2}$ | $5.04 \times 10^{-3}$ | $\lambda_{opt} = 1.25 \times 10^{-1}$ $\beta_{opt} = 29.67$ |

## S5    Details and Additional Results on 1D Reaction-Diffusion Equation

### S5.1    Details of Experiment Setup

The reaction-diffusion equation is specified as:

$$\frac{\partial u}{\partial t} - \nu \frac{\partial^2 u}{\partial x^2} - \rho u(1 - u) = 0 \tag{5}$$

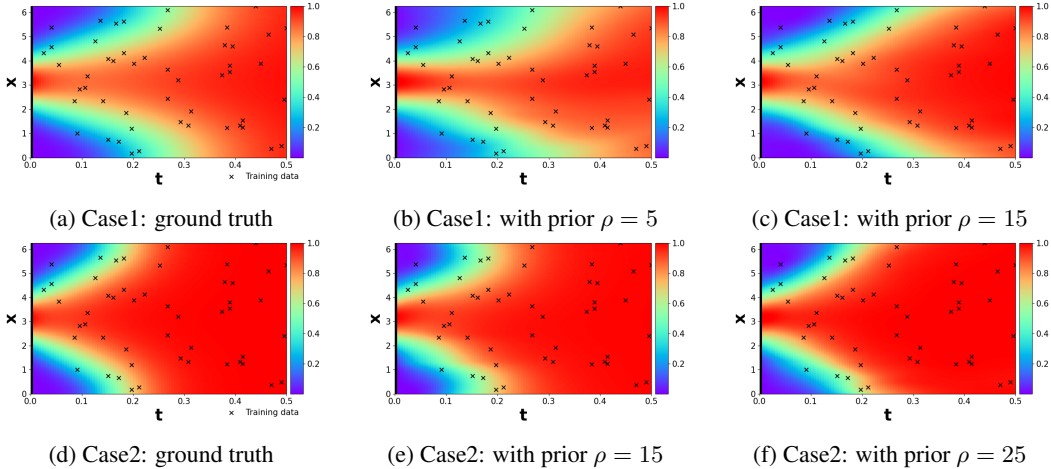

(a) Case1: ground truth      (b) Case1: with prior $\rho = 5$      (c) Case1: with prior $\rho = 15$

(d) Case2: ground truth      (e) Case2: with prior $\rho = 15$      (f) Case2: with prior $\rho = 25$

Figure S11: Two cases of 1D reaction-diffusion equation, including the ground truth and models with different physics priors. The crosses indicate the collocation points of training data.

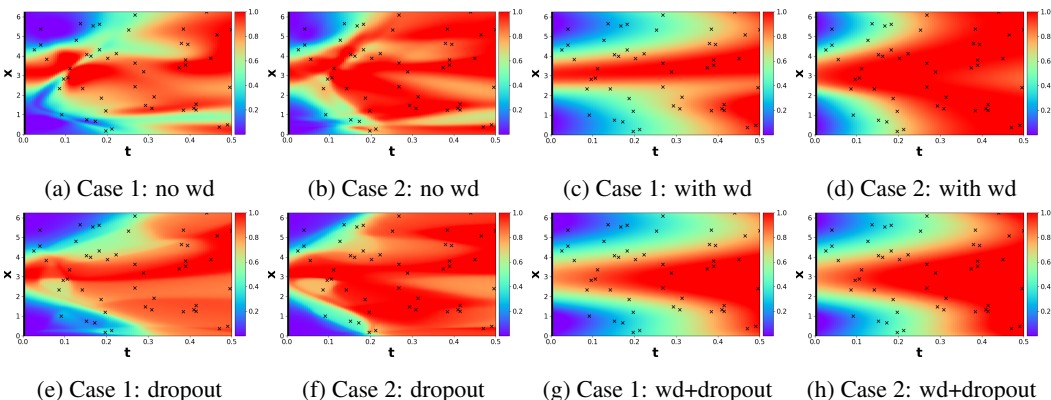

(a) Case 1: no wd    (b) Case 2: no wd    (c) Case 1: with wd    (d) Case 2: with wd

(e) Case 1: dropout    (f) Case 2: dropout    (g) Case 1: wd+dropout    (h) Case 2: wd+dropout

Figure S12: Baseline performance for reaction-diffusion without regularization, with weight decay, with dropout, and with dropout+wd.

with the associated initial and (periodic) boundary conditions $u(x, 0) = g(x)$, $x \in D$ and $u(0, t) = u(2\pi, t)$, $t \in (0, T]$ respectively, with $g(x) = \exp\left(-\frac{(x-\pi)^2}{2(\pi/4)^2}\right)$, with $\rho$ and $\nu$ being the reaction and diffusion coefficients respectively. Here, we fix $\nu = 3$ for all the experiments. We generate two datasets by using two oracle reaction-diffusion equations, shown in Fig. S11a and S11d respectively. Fig. S11 also shows results of learned models with different physics priors, all are different from the oracle models.

As a comparison, we also train the baseline models for the two reaction-diffusion equation cases from observation data only, with/without using weight decay as regularization. The results are shown in Fig. S12. Quantitative MSE from testing are tabulated in Tab. S7. Clearly our method outperforms weigh decay in all but one case.

**Data generation.** We generate data points using "oracle" models which are also reaction equations. For each case, we consider a mesh which consists of 100 time points between $T = [0, 0.5]$, with 256 spatial points at each time point. This results in a total of 25,600 grid points. All 256 spatial points for $T = 0$ are included in the loss function term related to the initial condition in Eqn. 1. Therefore, we have $([\hat{x}_j^I, 0], \hat{y}_j^I)$, where $\hat{y}_j^I = u(\hat{x}_j^I, 0)$ for $j = 1, 2, \ldots, N_I$, and $N_I = 256$. Similarly, we compute the boundary loss for the periodic boundary condition using Eqn. (2) for the boundary time points $[0, \hat{t}_1^b], [0, \hat{t}_2^b], \ldots, [0, \hat{t}_{N_b}^b]$, with $N_b = 100$. Additional 50 data points are randomly chosen to be included in the loss function $\mathcal{L}_d$, with $\mathcal{N}(0, 0.1)$ noise added to the them. Two "oracle" models are considered:

Table S7: Testing MSE for 1D Reaction-Diffusion Equation

| | prior used | baseline w/o reg. | baseline w/ weight dacay | $\lambda = \lambda_{opt}$ | |
|---|---|---|---|---|---|
| case 1 | $\rho = 5$ $\rho = 15$ | $1.60 \times 10^{-2}$ | $9.74 \times 10^{-3}$ | $2.68 \times 10^{-3}$ $1.33 \times 10^{-3}$ | $\lambda_{opt} = 1.12 \times 10^{-3}$ $\lambda_{opt} = 5.77 \times 10^{-3}$ |
| case 2 | $\rho = 15$ $\rho = 25$ | $1.33 \times 10^{-2}$ | $4.07 \times 10^{-3}$ | $9.22 \times 10^{-4}$ $6.33 \times 10^{-4}$ | $\lambda_{opt} = 9.67 \times 10^{-2}$ $\lambda_{opt} = 3.95 \times 10^{-3}$ |

Table S8: Testing MSE for 1D Reaction-Diffusion Equation

| | prior used | baseline w/ dropout | baseline w/ wd+dropout | $\lambda = \lambda_{opt}$ | |
|---|---|---|---|---|---|
| case 1 | $\rho = 5$ $\rho = 15$ | $5.59 \times 10^{-3}$ | $1.68 \times 10^{-2}$ | $2.68 \times 10^{-3}$ $1.33 \times 10^{-3}$ | $\lambda_{opt} = 1.12 \times 10^{-3}$ $\lambda_{opt} = 5.77 \times 10^{-3}$ |
| case 2 | $\rho = 15$ $\rho = 25$ | $7.70 \times 10^{-3}$ | $3.38 \times 10^{-2}$ | $9.22 \times 10^{-4}$ $6.33 \times 10^{-4}$ | $\lambda_{opt} = 9.67 \times 10^{-2}$ $\lambda_{opt} = 3.95 \times 10^{-3}$ |

- Case 1: $\rho = 10$
- Case 2: $\rho = 20$

We want to emphasize that the oracle models are never used as the physics prior.

Finally we randomly choose 100 collocation points in the support to compute generalized regularizers based on the physics priors. To provide quantitative measure on the accuracy of the learned model, we use all remaining points to compute mean square error (MSE) of the trained models.

### S5.2 Experiments with Multiple Physics Priors and Co-optimization of Physics Coefficients

In addition, we use Case 2 of the reaction-diffusion equation experiment for demonstration of multiple physics priors and co-optimization of physics prior coefficients, as shown in Fig. S13. In the first experiment, two physics priors are used, which are specified with $\rho = 15$ and $\rho = 25$. The optimal regularization parameters are $\lambda_1^* = 3.63 \times 10^{-2}$ and $\lambda_2^* = 8.19 \times 10^{-4}$ with the testing MSE of $8.43 \times 10^{-4}$. Also note the testing MSE is smaller than the individual cases when only one physics prior is used, as shown in Tab. S7.

In the second experiment, the final physics prior coefficient after optimization is $\rho^* = 19.06$. With an optimal $\lambda^* = 7.75 \times 10^{-2}$, it achieves the testing MSE of $4.37 \times 10^{-4}$. See Table S9 for details.

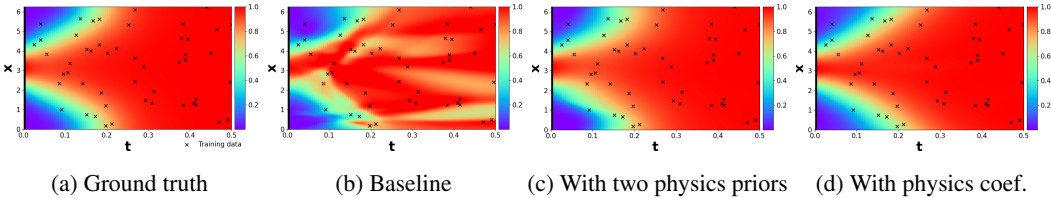

(a) Ground truth      (b) Baseline      (c) With two physics priors      (d) With physics coef.

Figure S13: Performance heatmap of case 2 of reaction-diffusion equation. (a) represents the ground truth, (b) shows the performance of baseline model (without weight decay), (c) shows results with two physics priors, while (d) with co-optimization of the physics coefficient $\rho$.

## S6   Repeatability Study

As a repeatability study, we choose Case 2 in the reaction experiment, and Case 1 in the convection experiment. In each case, we repeated baseline training (with and without weight decay) and our

Table S9: Testing MSE for Case2 of 1D Reaction-Diffusion Equation: Multiple priors and Physics coefficients. The last column indicates the optimal $\lambda$ values, and the physics coefficients, respectively.

|  | baseline w/o reg. | baseline w/ weight decay. | $\lambda = \lambda_{opt}$ |  |
|---|---|---|---|---|
| Multiple priors | $1.33 \times 10^{-2}$ | $4.07 \times 10^{-3}$ | $8.43 \times 10^{-4}$ | $\lambda_{opt_1} = 3.63 \times 10^{-2}$ (Prior 1: $\rho = 15$) $\lambda_{opt_2} = 8.19 \times 10^{-4}$ (Prior 2: $\rho = 25$) |
| Optimizing phys coeffs. | $1.33 \times 10^{-2}$ | $4.07 \times 10^{-3}$ | $4.37 \times 10^{-4}$ | $\lambda_{opt} = 7.75 \times 10^{-2}$ $\rho_{opt} = 19.06$ |

method three times. The testing MSE are reported in Tab. S10 and Tab. S11 respectively. The first value is the mean of the testing MSE, while the number in parenthesis is the standard deviation.

Table S10: Repeatability studies for Case 2 in Reaction Experiment. The first number is the average testing MSE of three runs with random seeds, while the number in parenthesis is the standard deviation.

| Oracle | Prior | Baseline | Weight decay | $\lambda = \lambda_{opt}$ |
|---|---|---|---|---|
| case 2 | $\rho = 15$ | $1.62 \times 10^{-2}$ ($2.72 \times 10^{-3}$) | $3.34 \times 10^{-3}$ ($4.81 \times 10^{-4}$) | $\mathbf{1.53 \times 10^{-3}}$ ($6.59 \times 10^{-4}$) |
|  | $\rho = 25$ |  |  | $\mathbf{2.50 \times 10^{-3}}$ ($5.34 \times 10^{-4}$) |

Table S11: Repeatability studies for Case 1 in Convection Experiment. The first value is the average testing MSE while the value in parenthesis is the standard deviation.

| Oracle | Prior | Baseline | Weight decay | $\lambda = \lambda_{opt}$ |
|---|---|---|---|---|
| Case 1 | $\beta = 25$ | $5.32 \times 10^{-2}$ ($2.31 \times 10^{-2}$) | $3.61 \times 10^{-2}$ ($6.53 \times 10^{-3}$) | $\mathbf{1.20 \times 10^{-2}}$ ($2.56 \times 10^{-3}$) |
|  | $\beta = 35$ |  |  | $\mathbf{9.82 \times 10^{-3}}$ ($8.87 \times 10^{-4}$) |

