# OpenReview forum: "Deep Learning with Physics Priors as Generalized Regularizers"
_NeurIPS.cc/2023/Workshop/AI4Science — NeurIPS2023-AI4Science Poster_

### Official Review · Reviewer_rVWW · 2023-10-12
**Good submission that needs some polishing for the camera-ready version**

**Rating:** 7
**Confidence:** 3

**Review:**

SUMMARY:

This paper discusses the regularizing neural networks with physics priors. This is done by including additional collocation points in the support of the model on which the prior is evaluated. The approach is evaluated on Hamilton’s equations, an 1D reaction equation and an 1D convection equations. It is shown that the proposed regularizer improves performance in most of the cases.

STRENGTHS:

- Experiments are extensive and the results look promising.

WEAKNESSES:
- To me the contributions of the paper are not very clear. Incorporating physics priors into model training is not new (however, I am not very familiar with this area of research.)
- The abstract is very vague and somewhat short. I suggest the authors improve/extend it for the camera-ready version.
- Sections 1 and 2 are hard to read. Here the writing could be improved to make more clear what the authors are doing. What exactly is the task the authors are trying to solve? Authors could improve the explanation of the physics prior (Section 2), e.g., by explicitly define it in a clearer way.
- Also, the mathematical notation can be improved at some places:
	1. Either in Equation (1) is missing the L2-norm or the observations are scalar (which is not properly defined).
	2. Equation (7) is not properly included in a sentence.
	3. Equation (14) and (15) could be improved by using the `\underbrace` command.
- Table 1 should also emphasize results where the regularized model does not perform better!
- Also it would be useful when a link to the used code repository by Geydanus could be provided.
- Overall the paper could be more polished, e.g., there are white spaces missing in front of references.

MINOR:
- In Equation 6 the lambda could be put in the nominator of the fraction.
- Missing white space in line 34


CONCLUSION:

Overall, the submission would make a nice contribution to the workshop and recommend to accept it. However, the authors should spend some more time to polish the paper for the camera-ready submission.

---

### Official Review · Reviewer_qYom · 2023-10-24
**Review of "Deep Learning with Physics Priors as Generalized Regularizers"**

**Rating:** 7
**Confidence:** 4

**Review:**

This work proposes a method for incorporating approximate models as physics priors in model training. This is done by structuring the priors as generalized regularizers, which is shown to significantly improve testing accuracy. This is particularly useful for situations in which there exists an approximate model of the physical system that we would like to try and incorporate into a machine learning model.

In particular, they use Vapnik's structural risk minimization (SRM) as the inductive principle to cast the generalized regularization as an optimization problem. By structuring the mechanistic model as a generalized regularizer, they are able to incorporate information from the approximate model into the model training.

They present experimental results based on their method for three different physical systems (ideal / real pendulum, 1D reaction equation, and a 1D convection example). In all but one case, their analysis suggests that their approach improves test accuracy compared to the baseline model.

Overall, this is a very interesting approach that attempts to tackle the longstanding problem of incorporating world knowledge into machine learning models. The authors do so by reframing physics priors as generalized regularizers and applying Vapnik's structural risk minimization inductive principle to balance complexity vs. accuracy of the resulting model.

- Minor issues:
  - [26]: ~~physics odel~~ --> physics **m**odel
  - [27]: ~~simplied~~ --> sim**p**lified
  - [71]: In the case the physics prior is an ODE,
    - maybe:
      > In the case **where** the physics prior is an ODE
      ?

---

### Meta-Review · Area_Chair_6H5a · 2023-10-27

**Recommendation:** Accept (Oral)
**Confidence:** 4

**Metareview:**

This work incorporates physics priors as generalized regularizers when training neural networks.
The paper reads well, though it contains some flaws, highlighted by the referees, which I urge the authors to fix for the camera-ready version.

Overall, the theory is well presented alongside good mathematical details.
There's an extensive experiments section that supports the theoretical arguments previously introduced.

Overall, this is a good paper that holds promises for exciting applications for this method. Given the originality of the work, upon a careful polishing as suggested by the referees, I recommend acceptance of this paper either as a poster or as an oral.